# Comparative genomics of *Lentilactobacillus buchneri* reveals strain-level hyperdiversity and broad-spectrum CRISPR immunity against human and livestock gut phages

Ismail Gumustop[1], Ibrahim Genel[2], Ibrahim Cagri Kurt[1]*, Fatih Ortakci[3]*

**1** Department of Molecular Biology and Genetics, Faculty of Arts and Sciences, Bogazici University, Istanbul, Turkiye, **2** School of Medicine, Koc University, Istanbul, Turkiye, **3** Department of Food Engineering, Faculty of Chemical and Metallurgical Engineering, Istanbul Technical University, Istanbul, Turkiye

* ibrahim.kurt@bogazici.edu.tr (ICK); ortakci@itu.edu.tr (FO)

## Abstract

This study conducted a comparative genomic investigation of 40 strains of *Lentilactobacillus buchneri* isolated from various environments—including fermented foods, silage, cattle rumen, and the nasopharynx—to identify species-level diversity and assess their CRISPR immunity. An average genome size of $2.55 \pm 0.07$ Mb, a GC content of $44.18 \pm 0.15\%$, and $2444 \pm 83$ coding sequences were identified. Prophages were found in all strains except for two, while 17 strains contained plasmids. No genes associated with bacteriocins were identified. CRISPR analysis revealed the presence of 42 Type II-A and 45 Type I-E systems, with each strain having at least one Type II-A system (~ 2 systems per strain). Among the 33 tested strains, 29 encoded complete LbCas9 proteins, consisting of 1371 amino acids. In-silico analysis of PAM in Type II-A systems revealed a 5'-DNAWDHV-3' motif, with a noted preference for 5'-AAAA-3' at positions 3–6. The spacers found in CRISPR arrays targeted proteins involved in plasmid mobilization as well as components of phage tails, indicating their roles in inhibiting horizontal gene transfer and providing defense against phages. Remarkably, 27 spacers from 24 strains were found to match phages associated with human gut microbiomes, with several showing the ability to cross-target phages from livestock, kefir, and wastewater. This research expands the genomic understanding of *L. buchneri* from 10 to 40 genomes, uncovering the dynamics of CRISPR-phage co-evolution. The defined PAM preferences of the identified CRISPR systems, together with the broad predicted target range of their spacers, highlight their potential for biotechnological applications—most notably targeted CRISPRization of *L. buchneri* strains and in-silico-guided phage control during fermentation. These findings deepen our understanding of the ecological adaptability of *L. buchneri* and provide a foundation for future industrial exploitation of its native CRISPR immunity.

**Data availability statement:** Genomes analyzed in the present study are available in NCBI with the following accession numbers: GCA_025186255.1, GCA_025186245.1, GCA_025186205.1, GCA_018314255.1, GCA_000298115.2, GCA_025194225.1, GCA_025194165.1, GCA_025194175.1, GCA_025194235.1, GCA_025194265.1, GCA_025194415.1, GCA_025194205.1, GCA_025212255.1, GCA_025190085.1, GCA_001434735.1, GCA_009495635.1, GCA_009495565.1, GCA_009495555.1, GCA_009495585.1, GCA_009495575.1, GCA_009495485.1, GCA_009495465.1, GCA_008369805.1, GCA_007992235.1, GCA_013167855.1, GCA_000211375.1, GCA_023507525.1, GCA_902797765.1, GCA_005048025.1, GCA_005049145.1, GCA_005047285.1, GCA_005048055.1, GCA_005047235.1, GCA_005049245.1, GCA_005047265.1, GCA_005047575.1, GCA_005049205.1, GCA_013167835.1, GCA_022651845.1.

**Funding:** This work has been supported by the Istanbul Technical University Scientific Research Projects Unit with grant number MGA-2023-45115. The funders had no role in study design, data collection and analysis, decision to publish, or preparation of the manuscript.

**Competing interests:** The authors have declared that no competing interests exist.

## Introduction

*Lentilactobacillus buchneri* (*L. buchneri*), named after German microbiologist E. Buchner, is an obligatory heterofermentative lactic acid bacteria species that was previously known as *Bacillus buchneri*. *L. buchneri* was repeatedly isolated from several types of hard and spreadable cheese, namely Canestrato Pugliese and Ricotta Forte, respectively. The reference strain for this species is *L. buchneri* ATCC 4005, which was isolated from tomato pulp [1,2].

Several similarities were observed between *Levilactobacillus brevis* and *Lentilactobacillus buchneri*, including their isolation from similar sources. Under light microscopy, *L. buchneri* appears as short chains of rods or single cells with rounded ends, measuring 0.7–1.0 × 2–4 µm. The species grows between 15 °C and 37 °C but fails to proliferate at 45 °C. Its genomic GC content ranges from 44% to 46%, and its peptidoglycan is of the lysine–D-aspartyl type. The capacity to ferment xylose, sucrose, lactose, galactose, raffinose, and esculin differs among strains. Two phenotypic traits that distinguish *L. buchneri* from *Levilactobacillus brevis* are its ability to ferment melezitose and the notably slow electrophoretic migration of its dual lactate dehydrogenase enzymes [1].

Fermented dairy foods are potential reservoirs for biogenic amines because of their rich amino acid content and possible metabolism of amino acids by adventitious non-starter lactic acid bacteria (LAB) during cheese making and ripening. Although the primary microbes associated with biogenic amine formation are *Enterobacteriaceae* and *Pseudomonas*, the main culprits for biogenic amine production are LAB that belong to *Lactobacilli*, *Lactococcus*, *Leuconostoc*, *Streptococcus*, and *Enterococcus*. Among *Lactobacilli*, the main biogenic amine formers are *L. buchneri*, *L. parabuchneri*, *Latilactobacillus curvatus*, *Lactobacillus helveticus*, *Limosilactobacillus vaginalis*, and *Levilactobacillus brevis*. It was reported that *L. buchneri* and *L. parabuchneri* can form histamine at refrigerator temperatures, but this phenotype can be strain-specific and depends on a complete histidine-decarboxylase (hdc) gene cluster [3,4].

*Lentilactobacillus buchneri* is an authorised and widely used starter culture for producing silage intended for animal feed. Its popularity stems largely from its ability to increase aerobic stability by limiting fungal proliferation. Homofermentative inoculants—such as *Lactobacillus acidophilus*, *Lactiplantibacillus plantarum*, *Pediococcus cerevisiae*, *Pediococcus acidilactici* and *Enterococcus faecium*—also enhance lactic acid production, yet silages fermented with these cultures can become less stable during aerobic storage than uninoculated material. This reduced stability has been linked to fungal metabolism of lactate when the product is exposed to oxygen. Because *L. buchneri* is heterofermentative, it converts part of the lactate into acetate, and the higher acetate concentration likely confers superior aerobic stability through its stronger inhibitory effect on yeasts and moulds [5].

Although *L. buchneri* inhabits diverse environments and is widely employed as a silage inoculant, its strain-level diversity has received only limited comparative genomic scrutiny. To close this gap, we analyzed every *L. buchneri* genome available in NCBI GenBank, examining both intra-species diversity and the CRISPR

immunome directed against human and livestock phages. Our in-silico survey covered 40 genomes—including the reference strain ATCC 4005—thereby expanding the dataset four-fold relative to the previous comparison of just ten strains [6].

## Methods

### Genome annotation

Whole-genome sequences of 46 *L. buchneri* strains were downloaded from NCBI with the following accession numbers of GCA_025186255.1, GCA_025186245.1, GCA_025186205.1, GCA_018314255.1, GCA_000298115.2, GCA_025194225.1, GCA_025194165.1, GCA_025194175.1, GCA_025194235.1, GCA_025194265.1, GCA_025194415.1, GCA_025194205.1, GCA_025212255.1, GCA_025190085.1, GCA_001434735.1, GCA_009495635.1, GCA_009495565.1, GCA_009495555.1, GCA_009495585.1, GCA_009495575.1, GCA_009495485.1, GCA_009495465.1, GCA_008369805.1, GCA_007992235.1, GCA_013167855.1, GCA_000211375.1, GCA_023507525.1, GCA_902797765.1, GCA_005048025.1, GCA_005049145.1, GCA_005047285.1, GCA_005048055.1, GCA_005047235.1, GCA_005049245.1, GCA_005047265.1, GCA_005047575.1, GCA_005049205.1, GCA_013167835.1, GCA_022651845.1.

Genome quality was assessed using BUSCO (version 5.4.3) [7] with lactobacillales_odb10 lineage dataset. Genomic annotation of sequences was accomplished using Prokka (version 1.14.5) with the following flag: --kingdom Bacteria.

### Comparative genomics of *L. buchneri*

The evaluation of genomic similarity across forty *L. buchneri* genomes was conducted by using Jaccard distance using the prabclus package [8] according to the absence or presence of genes. The resulting data were processed through Principal Coordinate Analysis (PCoA) using R (version 4.1.1) [9,10] to explore the relatedness among genomes.

The output of GFF files was analyzed using Roary (version 3.13.0) [11] with flags "-e -n -v -r," which allowed us to analyze the pan- and core genomes of the bacteria and compare the presence or absence of specific genes, including *hdcA* and *hisS*. The lower-end BLASTp identity was set to 95% in Roary to secure accuracy. To explore whether the *L. buchneri* pangenome is open, we used the micropan package [12] with 10,000 permutations to fit Heap's law model. These computational methods allowed us to get perception against the genetic makeup of *L. buchneri* and the variations of its pangenome.

The Single Nucleotide Polymorphisms (SNP) that exist in the core genome were identified using the snippy tool [13]. In addition, a core genome SNPs-based phylogenetic tree was constructed using Clustal Omega [14,15] and iTOL [16]. We calculated average nucleotide identity (CDS ANI) using the GET_HOMOLOGUES [17]. Genomic islands were identified using GIPSy [18] by feeding GenBank annotation files from Prokka. The genomes were aligned and visualized using the BLAST Ring Image Generator (BRIG) software with DSM 20001 as the reference genome, using the BLASTn algorithm with a lower identity threshold of 70% and a higher identity threshold of 90% [18,19].

We used the dbCAN2 meta server to annotate CAZymes (Carbohydrate Active Enzymes) encompassed by *L. buchneri* genomes. The CAZy annotation database (v11) was obtained after which HMMER (version 3.1b2) was applied for annotation of CAZyme domains [20,21]. The CAZyme annotation outputs were filtered by default coverage thresholds and e-values by using dbCAN2. Next, CAZyme families were utilized to categorize the *L. buchneri* isolates. To identify putative bacteriocin biosynthesis-related genes, we used BAGEL4 [22]. Plasmids were detected by using PLSDB [23,24]. Prophages were identified using PHASTEST [25] and GIPSy [18], respectively.

### CRISPR

CRISPRviz [26] was used to identify and visualize CRISPR loci in 40 strains. Types of loci are determined by CRISPR-CasFinder [27] and CRISPRClassify [28]. Open reading frames within +/-20kb of each array are predicted with Prodigal

v2.6.3 [29]. Hmmsearch function of HMMER v3.3.2 [21] (HMM alignment coverage >70%, bitscore >= 40, E-value < 1e-5) was used to annotate Cas proteins with curated HMM profiles accessed from [30]; proteins predicted to be partial by Prodigal [29] were removed during this process.

To analyze the relationship between diverse Cas9s, proteins named Cas9 (marked as non-fragment and longer than 800 aa) were downloaded from UniProt [31], sequences of commonly used Cas9s, and 33 Cas9s from 40 strains were used for multiple sequence alignment with MAFFT v7.526 [32]. The phylogenetic tree of this alignment was generated with FastTree v2.1.11 [33]. While only focusing on the phylogenetic relationship between 33 Cas9s from 40 strains, the same tools [32,33] and CRISPRviz [26] spacer and repeat results were used. To identify targets of the spacers, blastn [34] results of PAMPredict v1.0.2 [35], retrieved from searches against IMG/VR v4 [36] and IMG/PR [37], were examined. Megablast [34] search against the NCBI nt database [38] was performed to find out the top CDS hits of spacers. To predict the respective PAMs of 2 protein clusters, unique spacers belonging to the same clusters were gathered and provided as input to CRISPRUtils [39].

## Results

### Genomic characteristics

Forty-six *L. buchneri* genomes were downloaded from NCBI GenBank. All genomes were subjected to BUSCO [7] analysis to determine the completeness of the genomes (S1 Fig). Five genomes (CIRM-BIA 2085, UW_DM_LENLAC1_1, UW_DM_LENLAC1_2, UW_DM_LENLAC1_3, and UW_DM_LENLAC1_4) were excluded from further analysis due to showing below the threshold level (98%) completeness. The ATCC 11577 genome was discarded due to the incorrect taxonomic assignment of the genome assembly. Table 1 represents forty *L. buchneri* strains isolated from different ecologies such as silage, fermented dough, grape must, pickles, cheese, kimchi, ethanol production facility, cattle rumen, and clinical samples. Whole-genome sequence statistics of *L. buchneri* isolates showed that the average genome size was 2.55 ± 0.08 Mb, GC content was 44.18 ± 0.19%, and the number of CDS was 2443 ± 84.27, which are in alignment with the reference strain ATCC 4005.

### Comparative genomics

Forty *L. buchneri* genomes were processed through comparative genomic analysis to determine strain-level uniqueness and CRISPR immunome traits in this species. Forty genomes, including the reference strain ATCC 4005, were picked to construct a phylogenomic tree by using nucleotide sequence alignment of core genome -based single-nucleotide polymorphism (SNP) values of each individual strain (Fig 1). Two major clades were identified according to SNP-based phylogenetic evaluation. The first major clade from top-down mainly consisted of silage isolates and bovine nasopharynx. The exception to the first major clade was grape must that laid on in between two distinct ecological conditions. The second major clade originated from fermented cucumber slurry spoilage, fermented pickles, bovine nasopharynx, cattle rumen, cheese, fuel ethanol production facility, fermented vegetables, injera fermented dough, kimchi, and tomato pulp. Noteworthy, the reference strain ATCC 4005 that was isolated from tomato pulp clustered with another tomato pulp isolate of NBRC 107764. It was interesting to note that the bovine nasopharyn geal isolate of S51 excelled as an outlier strain that clustered with the second clade members.While silage isolates were primarily laid on the first major clade and closely located to each other on the phylogenetic tree, injera fermented dough isolates and artisanal fermented pickles were located in the second major clade. Moreover, fermented cucumber, fermented cucumber slurry spoilage, and fermented sorghum product isolates were closely located in the second major clade.

ANI (average nucleotide identity) based phylogenetic tree was constructed for quantitative analysis of similarity across forty *L. buchneri* strains (Fig 2). The ANI results showed that strains belonged to *L. buchneri* (ANI value > 96%). The ANI values calculated ranged from 97.41% to 99.99%. The minimum ANI reached was seen among LA1184 and CIRM-BIA 2081, which were isolated from reduced NaCl fermented cucumber spoilage and grape must, respectively. On the contrary, one of the maximum ANIs calculated was reached across the reference strain ATCC 4005 and DSM 20057, both of

**Table 1. Whole-genome assembly statistics of each of 40 *L. buchneri* genomes.**

| Strain | Isolation Source | Assembly | Sequencing Technology | Size (Mb) | GC% | CDS |
|---|---|---|---|---|---|---|
| 177 | Mais silage | GCA_025186255.1 | Illumina 2x125 | 2.52 | 44.2 | 2406 |
| 1012 | Injera fermented dough | GCA_025186245.1 | Illumina 2x125 | 2.56 | 44.1 | 2554 |
| 1014 | Injera fermented dough | GCA_025186205.1 | Illumina 2x125 | 2.47 | 44.3 | 2407 |
| ATCC 4005 | Tomato pulp | GCA_018314255.1 | PacBio RSII; Illumina HiSeq | 2.54 | 44.4 | 2414 |
| CD034 | Stable grass silage | GCA_000298115.2 | 454 | 2.56 | 44.2 | 2425 |
| CIRM-BIA 1514 | Silage | GCA_025194225.1 | Illumina 2x125 | 2.54 | 44.2 | 2433 |
| CIRM-BIA 1516 | Silage | GCA_025194165.1 | Illumina 2x125 | 2.58 | 44.1 | 2479 |
| CIRM-BIA 2081 | Grape must | GCA_025194175.1 | Illumina 2x125 | 2.63 | 44.0 | 2508 |
| CIRM-BIA 2082 | Artisanal fermented pickles | GCA_025194235.1 | Illumina 2x125 | 2.54 | 44.2 | 2440 |
| CIRM-BIA 2083 | Artisanal fermented pickles | GCA_025194265.1 | Illumina 2x125 | 2.52 | 44.3 | 2411 |
| CIRM-BIA 2084 | Artisanal fermented pickles | GCA_025194415.1 | Illumina 2x125 | 2.57 | 44.0 | 2473 |
| CIRM-BIA 659 | Tomato pulp | GCA_025194205.1 | Illumina 2x125 | 2.50 | 44.2 | 2402 |
| CIRM-BIA 664 | Conjunctiva | GCA_025212255.1 | Illumina 2x125 | 2.50 | 44.3 | 2403 |
| CIRM-BIA 845 | Cheese (domiati) | GCA_025190085.1 | Illumina 2x125 | 2.50 | 44.3 | 2399 |
| DSM 20057 | Tomato pulp | GCA_001434735.1 | Illumina MiSeq; Illumina HiSeq | 2.45 | 44.4 | 2356 |
| FUA3252 | A fermented sorghum product | GCA_031629455.1 | Oxford Nanopore MinION | 2.60 | 44.2 | 2483 |
| LA1147 | Reduced NaCl fermented cucumber spoilage | GCA_009495635.1 | Illumina HiSeq | 2.62 | 44.0 | 2508 |
| LA1161B | Anaerobic fermented cucumber slurry spoilage | GCA_009495565.1 | Illumina HiSeq | 2.62 | 43.9 | 2534 |
| LA1161C | Anaerobic fermented cucumber slurry spoilage | GCA_009495555.1 | Illumina HiSeq | 2.57 | 44.1 | 2470 |
| LA1167 | Anaerobic fermented cucumber slurry spoilage | GCA_009495585.1 | Illumina HiSeq | 2.62 | 44.0 | 2530 |
| LA1175D | Reduced NaCl fermented cucumber spoilage | GCA_009495575.1 | Illumina HiSeq | 2.68 | 44.0 | 2573 |
| LA1181 | Reduced NaCl fermented cucumber spoilage | GCA_009495485.1 | Illumina HiSeq | 2.63 | 44.0 | 2538 |
| LA1184 | Reduced NaCl fermented cucumber spoilage | GCA_009495465.1 | Illumina HiSeq and PacBio | 2.76 | 44.0 | 2690 |
| MGB0786 | Kimchi | GCA_008369805.1 | PacBio RSII | 2.58 | 44.3 | 2512 |
| MGR2–32 | Grass silage | GCA_033096565.1 | illumina NovaSeq; PromethION | 2.54 | 44.3 | 2393 |
| NBRC 107764 | Tomato pulp | GCA_007992235.1 | Illumina HiSeq 1000 | 2.49 | 44.2 | 2405 |
| NK01 | Silage | GCA_013167855.1 | Illumina MiSeq | 2.65 | 43.9 | 2513 |
| NRRL B-30929 | Fuel ethanol production facility | GCA_000211375.1 | Sanger/454/Illumina | 2.59 | 44.2 | 2504 |
| PC-C1 | Fermented vegetables | GCA_023507525.1 | Illumina | 2.53 | 44.2 | 2437 |
| RUG14303 | Cattle rumen | GCA_902797765.1 | NA | 2.33 | 44.6 | 2213 |
| S42 | Nasopharynx | GCA_005048025.1 | Illumina MiSeq | 2.50 | 44.3 | 2363 |
| S43 | Nasopharynx | GCA_005049145.1 | Illumina MiSeq | 2.50 | 44.3 | 2362 |
| S45 | Nasopharynx | GCA_005047285.1 | Illumina MiSeq | 2.49 | 44.3 | 2362 |
| S47 | Nasopharynx | GCA_005048055.1 | Illumina MiSeq | 2.45 | 44.4 | 2306 |
| S50 | Nasopharynx | GCA_005047235.1 | Illumina MiSeq | 2.54 | 44.1 | 2419 |
| S51 | Nasopharynx | GCA_005049245.1 | Illumina MiSeq | 2.54 | 44.2 | 2446 |
| S53 | Nasopharynx | GCA_005047265.1 | Illumina MiSeq | 2.50 | 44.3 | 2362 |
| S58 | Nasopharynx | GCA_005047575.1 | Illumina MiSeq | 2.50 | 44.3 | 2361 |
| S59 | Nasopharynx | GCA_005049205.1 | Illumina MiSeq | 2.54 | 44.1 | 2418 |
| SG162 | Silage | GCA_013167835.1 | Illumina MiSeq | 2.65 | 44.0 | 2533 |

which were isolated from tomato pulp. Notably, strains S42 through S59 were isolated in the same year (2014) and in the same location (Lethbridge, Canada). In addition, they demonstrate a very high level of intergenomic similarity (highest ANI values), with the exception of S51 (Fig 1, Fig 2). It is likely that they might be the isolates of one strain that went through genetic drift that are circulating in bovine nasopharynx in this particular farm microenvironment.

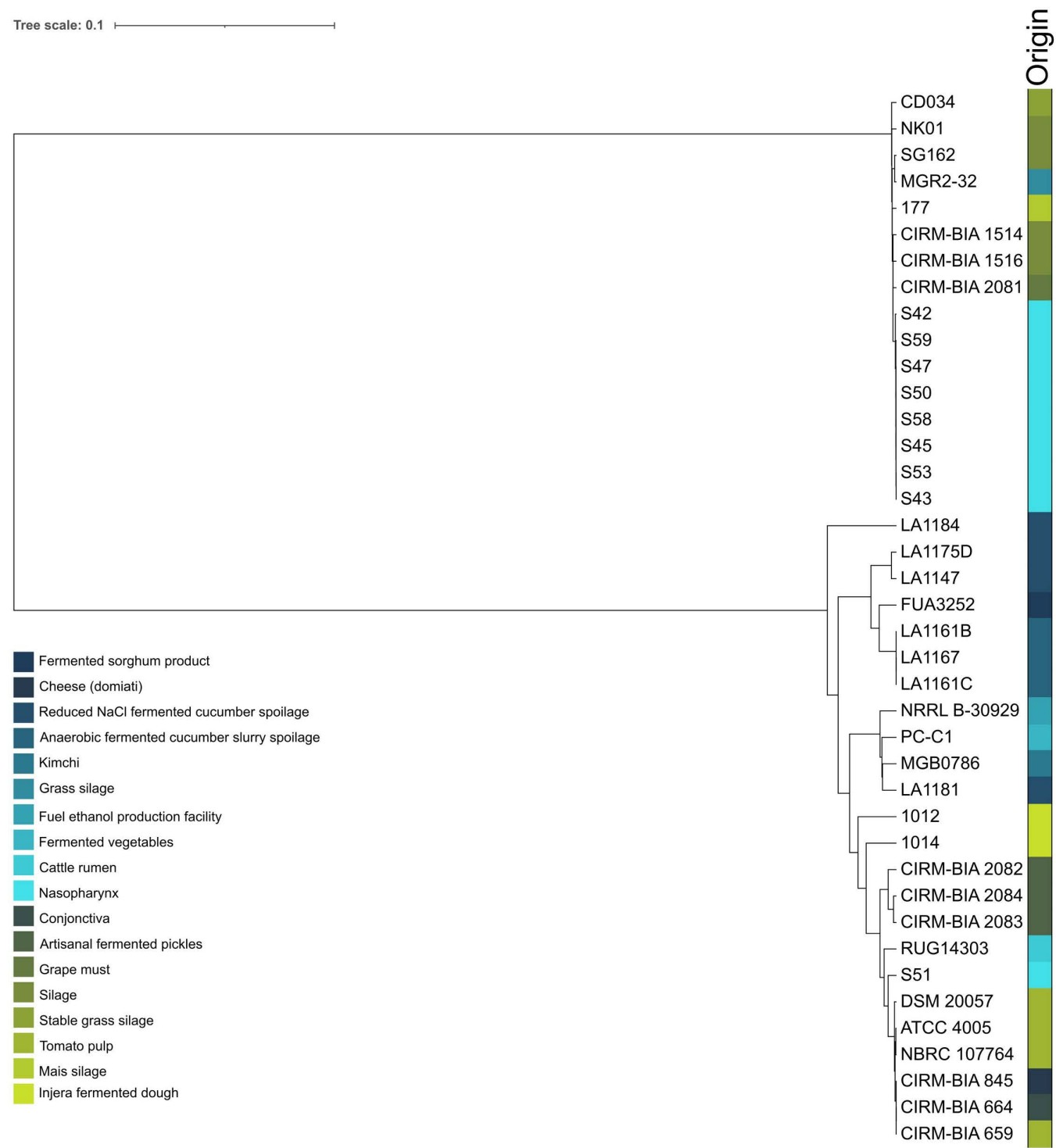

**Fig 1. SNP-based phylogenetic tree of 40 *L. buchneri* isolates and their isolation sources.**

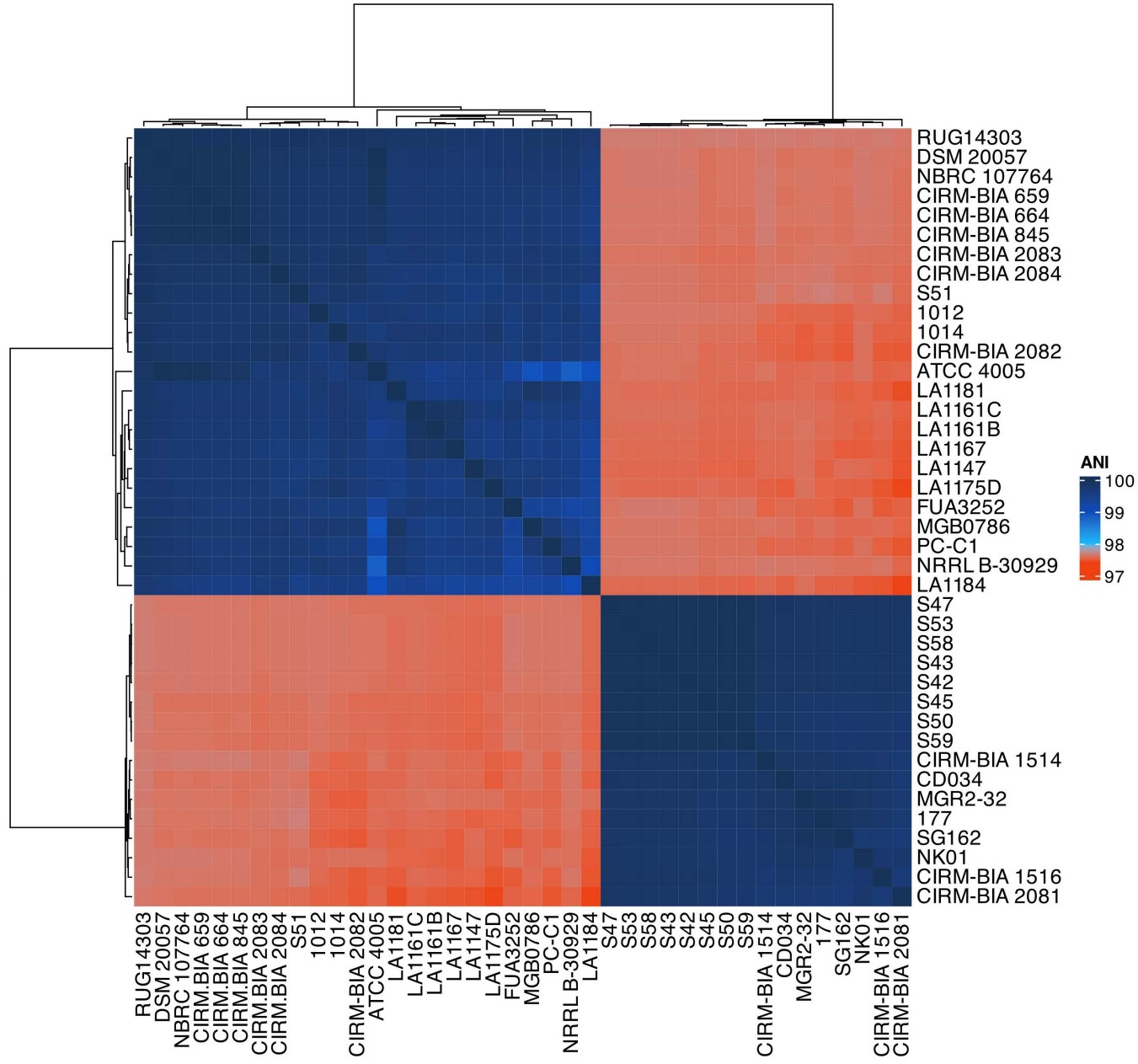

**Fig 2. Average nucleotide identity based heat map and clustering of 40 *L. buchneri* genomes.** The color gradient from red to dark blue shows a higher trend in percent identity.

We also constructed a PCoA plot based on Jaccard distance of presence and absence of genes existing in the pangenome that were annotated with Prokka, which is shown in Fig 3. A clear clustering of strains occurred on the negative and positive sides of PCo1. On the positive side of PCo1, 16 isolates were accumulated, half of which belonged to bovine nasopharynx and a quarter of which belonged to silage. On the negative side of PCo1, of the 24 strains, two-thirds of which belonged to fermented foods, including fermented cucumber, pickles, fermented dough, kimchi, and cheese. The rest of the strains belonged to tomato pulp, conjunctiva, cattle rumen, nasopharynx, and a bioethanol production facility.

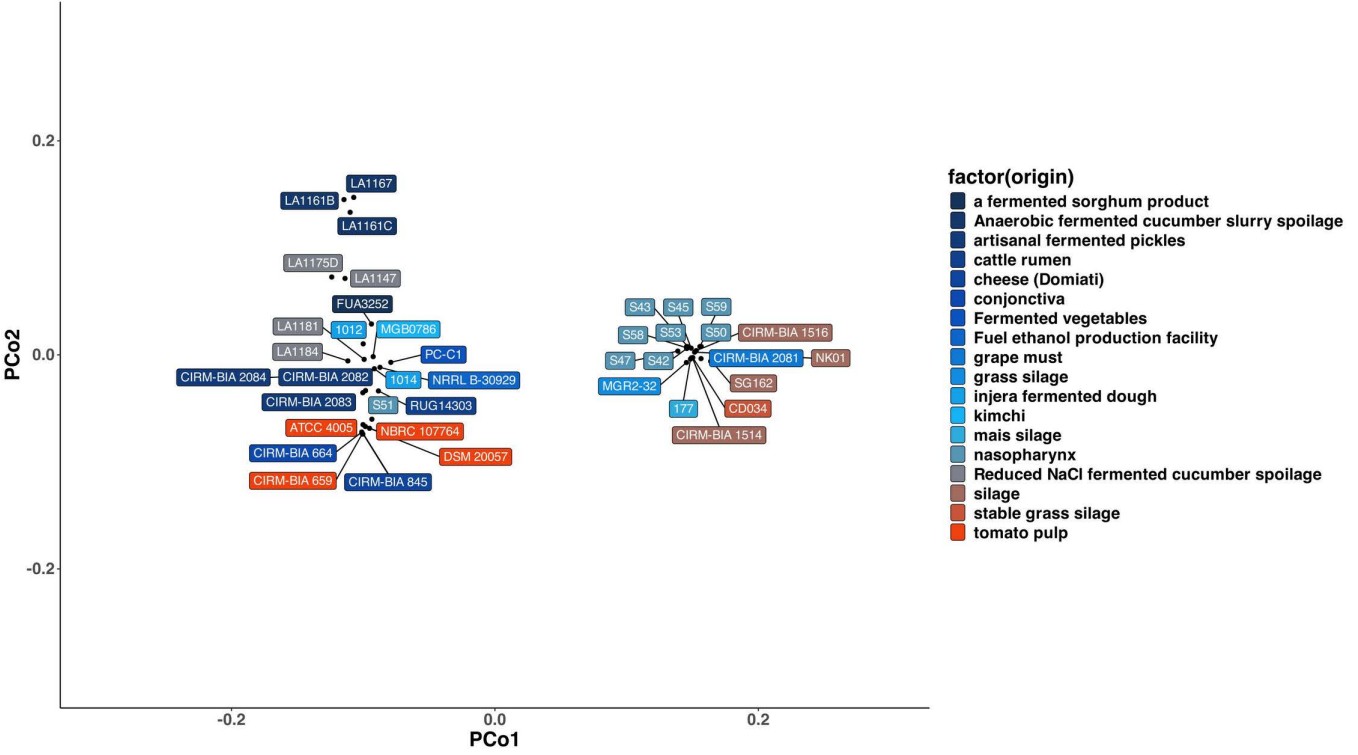

**Fig 3. PCoA graph of Jaccard distances based on shared genes among 40 *L. buchneri* genomes analyzed.** Each colored box shows a unique isolation source.

BLAST Ring Image Generator (BRIG) was used to run comparative whole genome analysis of forty *L. buchneri* genomes against the reference strain ATCC 4005 (Fig 4). Overall, comparison of putative coding sequences of all genomes versus reference strain ATCC 4005 revealed that a high identity percentage was seen with 70–100% identity as illustrated in Fig 4. Declining GC percentage and decreasing BLAST identity showed three major gaps in the BRIG image. The first gap in the coverage is between the 280–310 kb region, which contains a putative prophage. Similarly, the second and third gaps have a putative prophage region for each and are located at 1160–1200 kb and 2470–2510 kb regions, respectively.

### Core- and pangenome analysis

Analysis of genomic conservation across forty *L. buchneri* genomes resulted in 30.9% of the pangenome being conserved within the 95% BLASTP identity threshold which accounts for the core genome (Fig 5A). A total of 1006 genes, which corresponds to 17.3% of pangenome, forms shell genes whereas representation of cloud genes was achieved at 51.8% of total coding sequences suggesting phenotypic differences across *L. buchneri* strains analyzed [40]. To get a deeper understanding, random subsampling was implemented to construct each individual strain 's trendlines of core- and pange-nome (Fig 5B). The size of the core genome approached a plateau right around the 20th genome, whereas a flatline was not achieved with regard to the size of the pangenome. Because new genes are still in the process of being uncovered and the genome continues to increase, *L. buchneri* species has an open pangenome. Among the total genes identified in the pangenome, 1792 of them were common to all genomes, which represent the core genome. Across all *L. buchneri* genomes analyzed, 1012 was predicted to harbor 224 unique genes (Fig 5C). The second and third largest unique gene counts (i.e., 164 and 132) were achieved by MGB0786 and LA1184, respectively. On the contrary, *L. buchneri* S42 and S58 do not possess any unique genes compared to the other genomes screened.

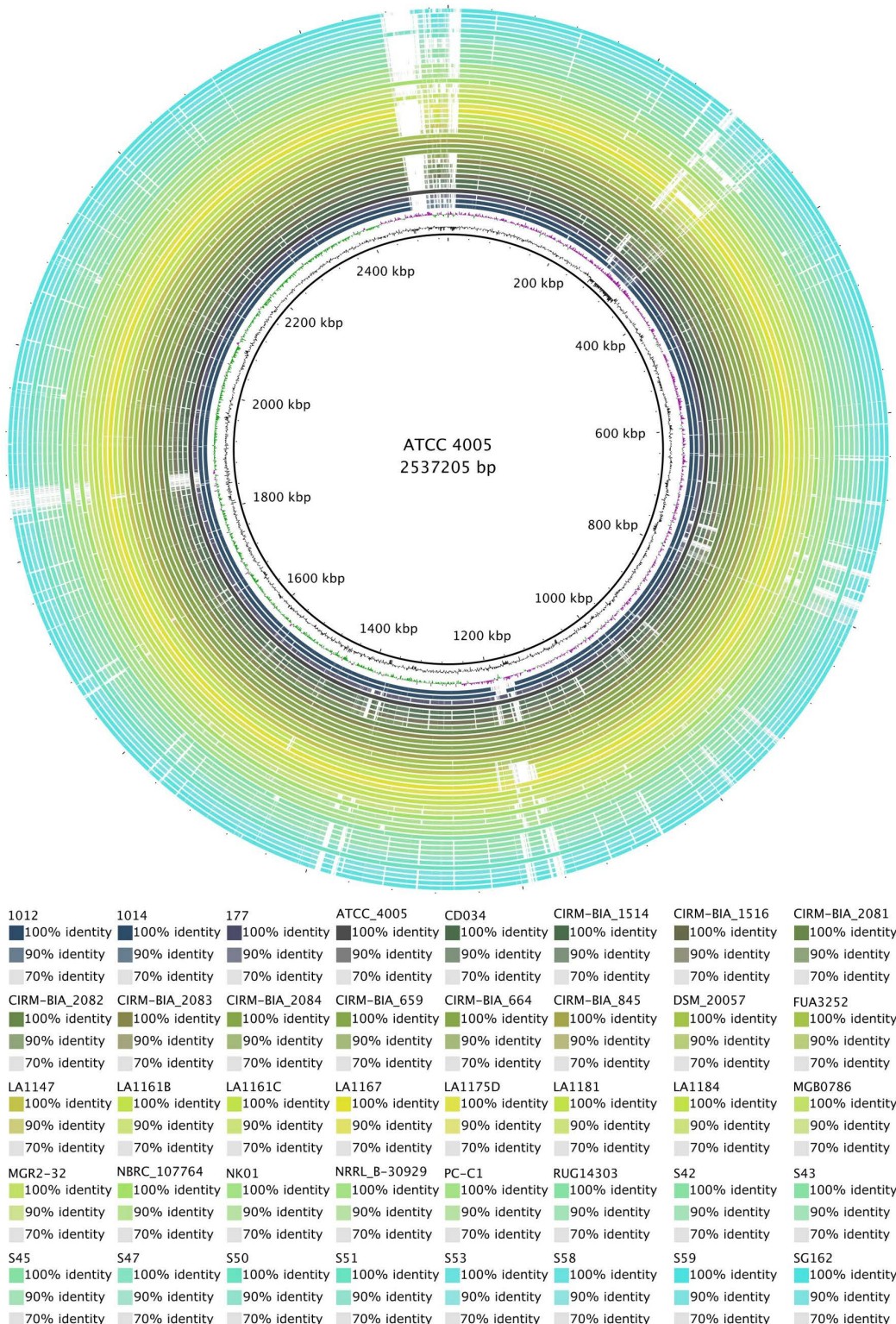

**Fig 4. BLAST Ring Image Generator analysis of 39 _L. buchneri_ genomes against reference genome ATCC 4005.** The innermost ring shows the location of the genome. Prophages and genomic islands were depicted outside of the rings.

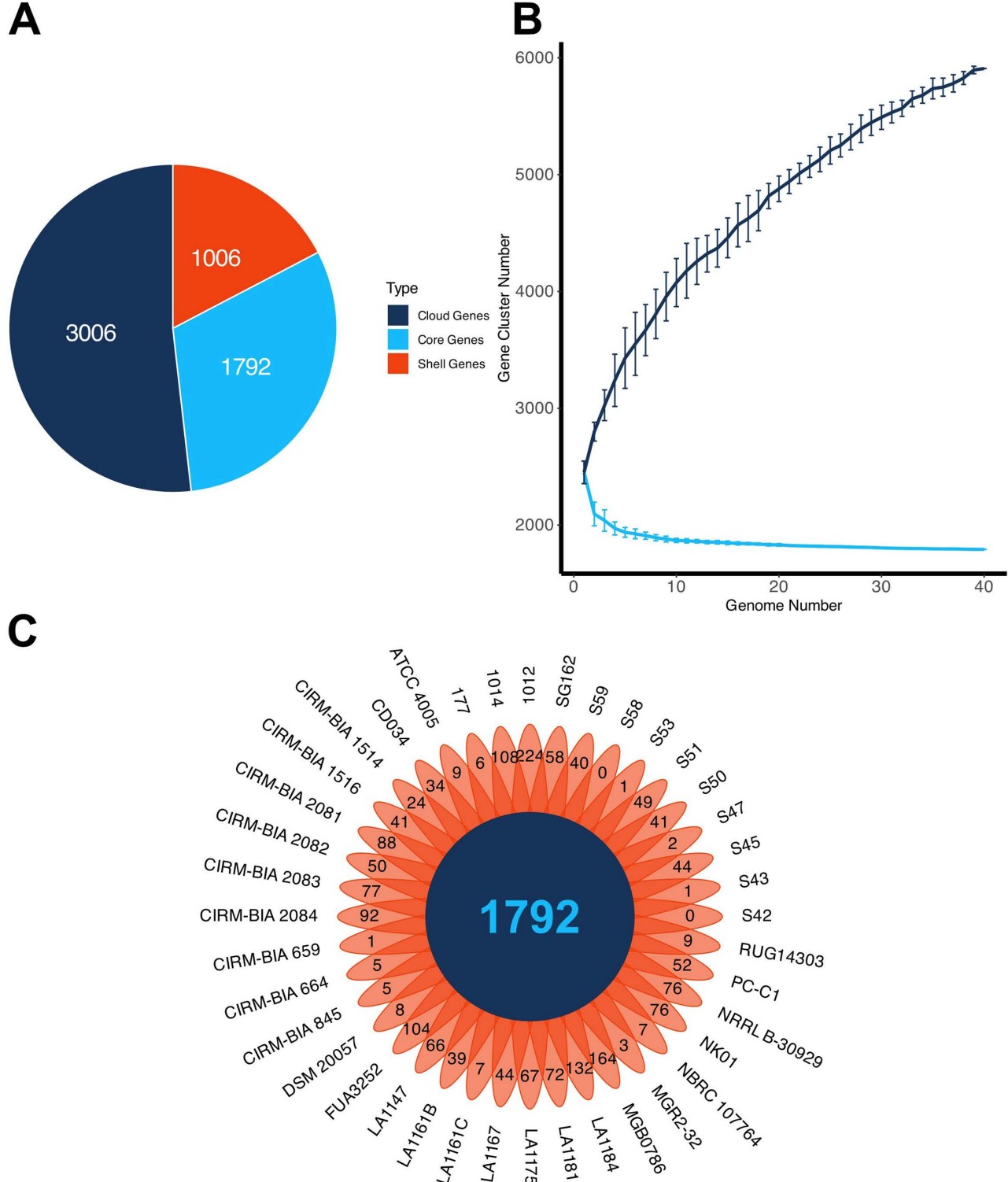

**Fig 5. (A) Distribution of the coding sequences across core and pangenome.** (B) Number of core genes (light green curve) vs pan genes (dark blue curve). (C) Flower plot of core and unique gene families of 40 *L. buchneri* isolates.

## Functional annotations

The core and pangenomes were annotated by utilizing eggNOG-Mapper [41,42]. Functional COG categories of orthogroups were appointed according to the Database of Clusters of Orthologous Genes [43]. The largest core- and pangenome categories were associated with genes with function unknown (Fig 6A). The second largest pangenome category was related to replication, recombination, and repair-associated genes. Amino acid transport and metabolism, along with translation, ribosomal structure, and biogenesis-related functional COG categories, form the second and third abundant genes. Transcription, as well as amino acid transport and metabolism, had similar pan and core genome-related genes as they pertained to functional COG categories. Cell motility, secondary metabolites biosynthesis, transport and catabolism had the least number of core- and pangenome associated genes.

Analysis of CAZymes indicated three major distinct clades according to the number of genes associated with each family of CAZymes (Fig 6B). The first clade from the very left was composed of six strains (CIRM-BIA 664, CIRM-BIA 659, CIRM-BIA 845, CIRM-BIA 2081, SG162, and NK01) that were isolated from conjunctiva, tomato pulp, cheese, grape must, rice grain silage, and silage, respectively. The second clade consisted of fifteen strains that were mostly isolated from either the conjunctiva, fermented vegetables, tomato pulp, fermented pickles, a fuel ethanol production facility, fermented cucumber spoilage, or silage. The third major clade contained 48% of the strains, including the reference strain ATCC 4005. The largest number of GH family enzymes encoding genes were discovered in SG162 isolated from rice grain silage. The second-largest GH encoding genes were detected in CIRM-BIA 2081, CIRM-BIA 659, CIRM-BIA 664, and CIRM-BIA 845. The third largest number of GH-related genes w as found in NK01 and MGR2–32. The highest number of GT encoding genes w as also found in NK01. CIRM-BIA 2084 ranked second according to the number of GT family enzyme-encoding genes. The CE, AA, and CBM content of individual *L. buchneri* was comparable to each other and remarkably smaller compared to that of GH and GT family CAZymes.

Genomic screening of all 40 *L. buchneri* strains for the existence of histidine gene clusters was performed based on the presence or absence of the genes of *hdcA* (histidine decarboxylase), *hdcB* (histidine decarboxylase maturation protein), *hisS* (histidine tRNA ligase), and *hdcC* (histidine/histamine antiporter). All strains were predicted to carry histidine tRNA ligase. LA1181, which was isolated from reduced NaCl fermented cucumber spoilage, was the only strain among 40 *L. buchneri* strains analyzed that was predicted to harbor a putative histidine decarboxylase gene. Still, the remaining genes in the *hdc* cluster (*hdcB* and *hdcC*) were not found in any of the strains tested.

## Mobile genetic elements

Forty *L. buchneri* isolates were explored for the existence of mobile genetic elements of prophages, plasmids, and the CRISPR-Cas system. Genome evaluations for the existence of prophages and plasmids detected 72 intact prophages (S2 Table) and 32 plasmids (S3 Table). Of the 40 *L. buchneri* genomes analyzed, 38 of those harbored at least one intact prophage and 17 were predicted to contain at least one plasmid. Of the genomes harboring prophages, 76% carried more than one intact element, and 41% of the plasmid-positive genomes encoded multiple putative plasmids. Strains 1014, LA1184, MGB0786, S51 and SG162 each contained the highest number of intact prophages—three per genome. The least number of prophages (1) were detected in CIRM-BIA 2082, CIRM-BIA 2083, CIRM-BIA 2084, LA1147, S42, S43, S47, S53, and S58. The bovine nasopharynx isolates overall had the highest number of intact prophages in their genome. This was followed by isolates of tomato pulp and anaerobic fermented cucumber slurry spoilage. No prophages were identified in LA1175D and RUG14303.

Across 21 unique plasmids detected, the NZ_CP073067.1 plasmid was the most abundant, which accounts for ~19% of all plasmids determined. The second most abundant plasmid was NZ_CP065817.1, which was identified four times. The following plasmids of NC_016035.1, LR962096.1, and NZ_CP043613.1 were identified twice. The rest of the plasmids were detected once only, and they account for half of the plasmids extracted. The highest number of plasmids w as

**A**

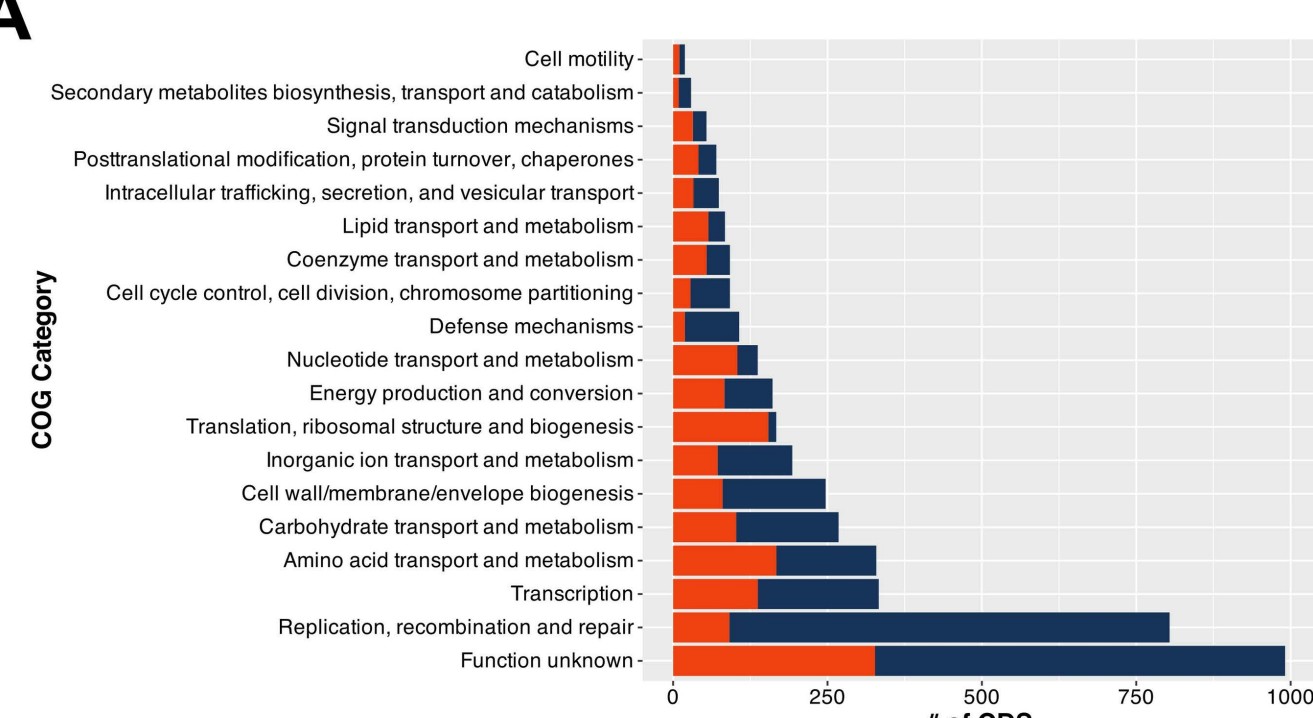

**B**

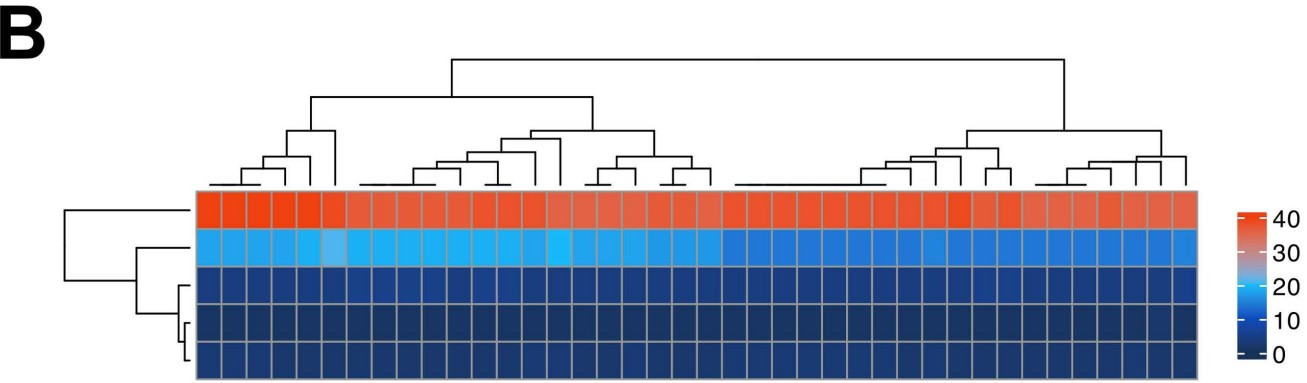

**Fig 6. (A) CAZyme heat map of 40 *L. buchneri* strains.** The color gradient from lighter to darker colors represent the abundance of CAZymes found in each genome. GH: Glycoside hydrolase, GT: Glycosyltransferase, CE: Carbohydrate esterase, AA: Auxiliary activity, CBM: Carbohydrate binding module. R programming language [10] (version 4.1.1) was used to draw the heatmap. (B) Functional COG analysis across core and pangenomes of 40 *L. buchneri* strains. (Core genes (Red), Accessory genes (Dark blue).

found in those strains isolated from reduced NaCl fermented cucumber spoilage and tomato pulp (6 and 4, respectively). No bacteriocin biosynthesis-related genes were found in any of the 40 *L. buchneri* strains screened.

## CRISPR-Cas systems

CRISPR-Cas systems enable bacteria and archaea to develop resistance against viruses, plasmids, and other foreign mobile genetic elements [44]. Because *L. buchneri* strains have potential applications in the food industry, computational identification and comprehensive characterization of all CRISPR loci in the 40 strains were also performed. First, 40

genomes were screened using PILER-CR for the presence of potential CRISPR loci to identify direct repeats and spacers [45]. Due to the fact that PILER-CR i) occasionally splits a single CRISPR array into multiple arrays, ii) outputs unusually short spacers (<10 nts) towards the ends of the arrays, iii) misses direct repeats with significant mismatches to the consensus direct repeat of the array, CRISPRviz tool was also employed in parallel [26]. CRISPRviz analysis of spacers revealed 11 main clades with multiple strains clustered and 9 unique clades with single members (S2A Fig). In contrast, CRISPRviz analysis for the direct repeats revealed 5 main clades with multiple strains and 5 unique clades with single members with occasional mutated repeats across the board in presumably historically older sequences at the ends of the arrays [26,46] (S2B Fig). CRISPR loci were then classified using two complementary approaches of CRISPRCasFinder and CRISPRClassify that are based on cas genes and direct repeats within loci, respectively [27]. This was conducted to capture any CRISPR loci that might be lacking cas genes or arrays [28].

While there was a significant overlap between the tools, screening of cas genes around the identified arrays using protein homology identified several CRISPR systems lacking cas effector proteins. Eighty-eight CRISPR loci carried an array with or without cas proteins, of which 84 were successfully assigned by CRISPRClassify (probability>0.90). Four loci with low prediction scores were then examined with C RISPRCasFinder. CrisprCasFinder was able to make a prediction for 3 of 4 loci, as one of the loci was missing cas proteins that are a prerequisite for CasFinder to function. In combination with both tools, 42 of Type II-A and 45 of Type I-E systems were characterized. Notably, all strains screened had carried at least one Type II-A system, with a mean of ~2 total CRISPR systems per strain (S1 Table).

As the most abundant CRISPR system found in *L. buchneri* strains and the most widely studied platform among all CRISPR types, we further characterized the effector proteins of the Type II-A systems, namely *L. buchneri* Cas9s (LbCas9). Twenty-nine out of 33 LbCas9s from 33 strains were putatively complete and 1371 amino acids in length. However, LbCas9s from S45, S47, FUA3252, CIRM-BIA_1514, and MGB0786 strains were truncated from their C termini and 644, 644, 1007, 1152, and 1284 amino acids in length, respectively. When compared to 3366 different Cas9s cataloged in Uniprot, LbCas9s clustered separately from Cas9 effectors that have been widely characterized and employed, such as SpCas9, SaCas9, CjCas9, and NmeCas9s (Fig 7A).

To elucidate whether the LbCas9 effectors of the members of the clades that formed based on direct repeat and spacer profiles previously were the same or similar, an unrooted phylogenetic analysis of the multiple protein sequence alignments of 33 LbCas9s from 33 strains w as performed. The phylogenetic tree revealed two main clades carrying 10 or 23 members, respectively (S3 Fig). The genetic distance between the furthest two LbCas9s was 4.96%, reflecting a close and conserved CRISPR relationship between the strains in alignment with the ANI values (Fig 2). While the direct repeat profiles did not completely segregate between the two LbCas9 clades, spacer profiles were clearly distinct (Fig 7B).

Next, characterization of 40 *L. buchneri* strains' CRISPR resistome was performed by focusing on the putative target protospacers of the spacers within our arrays with the goal of understanding the immune interactions among *L. buchneri*, bacteriophages, and other mobile genetic elements such as plasmids. PAMPredict tool was employed to search the spacers against 15,663,652 viral sequences from IMG/VR v4 and 699,973 plasmid sequences from IMG/PR [36,37]. The landscape revealed 30 clustered target profiles with diverse target organisms isolated from human and animal gut, sludge, kefir, anaerobic bioreactors, food waste, and wastewater microbial communities along with phages and plasmids from *Lactobacillus*, *Lentilactobacillus*, and *Agrilactobacillus* (Fig 8A). Comprehensive in-silico protospacer matching against viral and plasmid reference sets allowed us to infer the protospacer-adjacent motif (PAM) for the larger LbCas9 clade. Across the 29 strains in this group, nucleotide-logo analysis yielded a clear 5′-DNAWDHV-3′ consensus immediately downstream of the protospacer, with a pronounced 5′-AAAA-3′ signal at positions 3–6. The smaller Cas9 clade, however, contributed too few non-redundant protospacers to reach the statistical threshold required for reliable PAM calling. Notably, the adenine-rich signature matches the PAM previously reported for *L. buchneri* Cas9 in a smaller strain set, confirming that our larger dataset recapitulates and extends earlier findings [6] (Fig 8B).

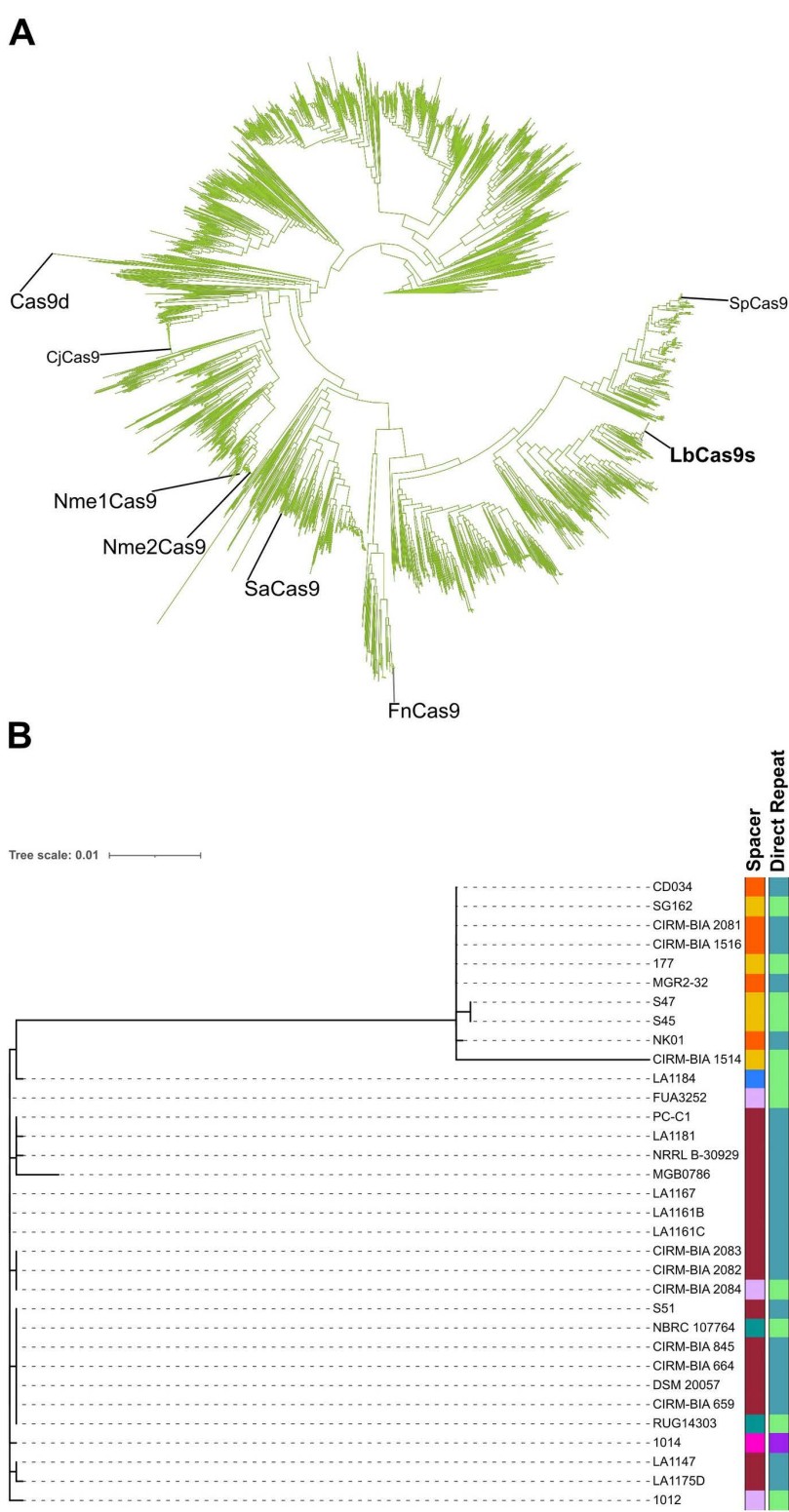

**Fig 7. (A) Phylogenetic tree of 3366 different Cas9s cataloged in Uniprot and 33 LbCas9s.** Widely characterized effectors and LbCas9s are denoted with additional lines. (B) Phylogenetic tree of 33 LbCas9s based on primary amino acid sequence similarity. Colors denote different groups identified by CRISPRviz regarding repeat and spacer similarity.

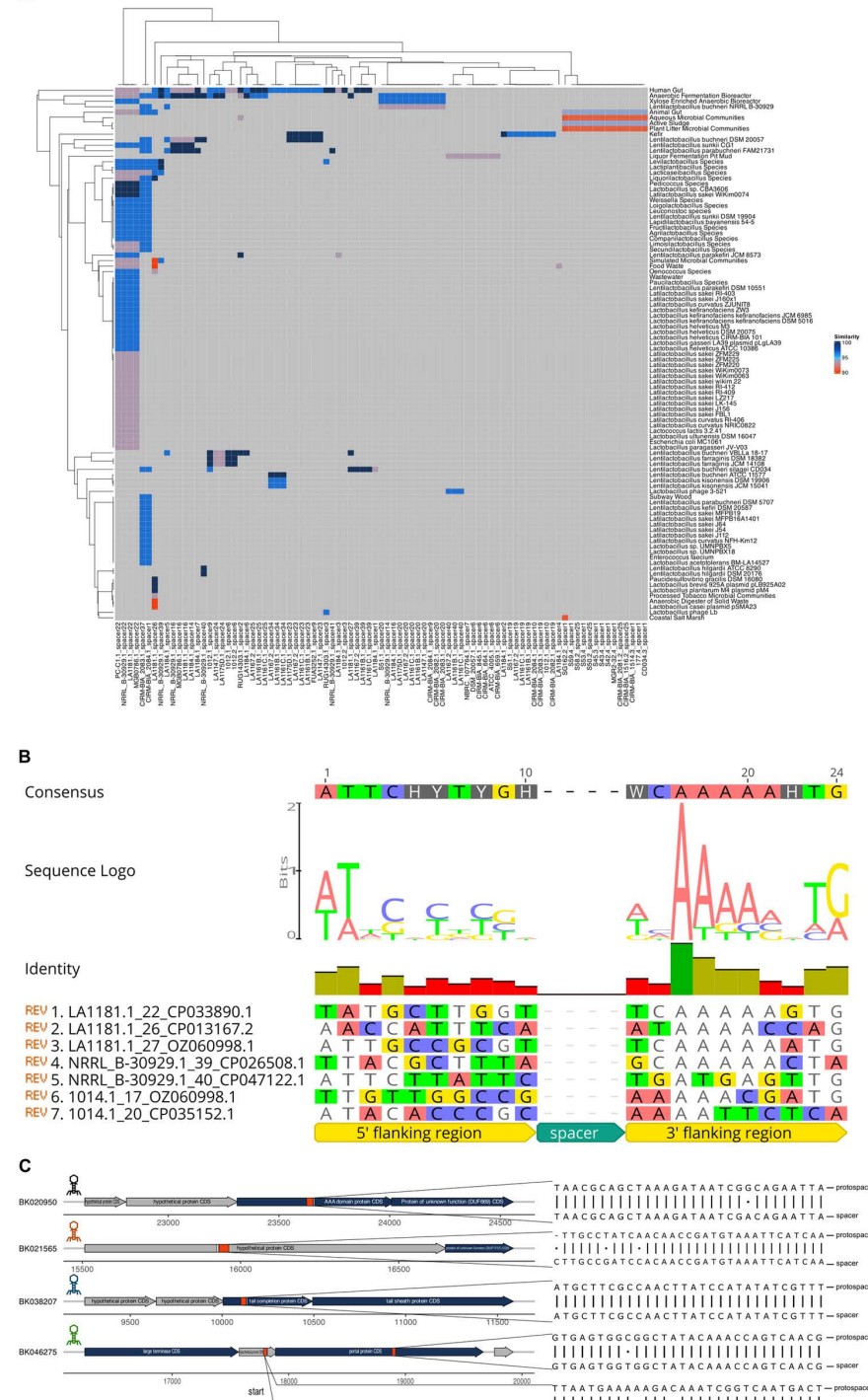

**Fig 8. (A) Heatmap of all spacers with significant hits, identified by PAMPredict, when searched against IMGVR and IMGPR databases.** Target samples are binned into categories instead of displaying the individual strain information. Dendrograms show the relationships within spacers and hits. (B) *In silico* predicted PAM sequence as determined by the alignment of 7 different protospacer-flanking sequences. (C) The genomic context of 4 distinct phages from human gut microbiome targeted by spacers are shown. In sequence alignment, the top strands represent the phage (protospacer) and bottom strands represent the spacer sequences.

To further characterize the coding DNA sequences (CDS) that are putatively targeted by the spacers, we employed a megablast search against NCBI's core nucleotide database [38,47]. This analysis corroborated the observation that CRISPR systems within *L. buchneri* could provide resistance against plasmids and bacteriophages (S4 Table). The target plasmid CDS elements were concentrated on the mobilization proteins that are crucial for plasmid transfers between bacteria, such as conjugal transfer protein (VirB4 of Type IV secretion system), mobA/mobL family protein, and mobilization protein 141 (mob141) [48–50] (S4 Table). The target viral CDS elements were mostly condensed on phage tail components along with other phage genes such as phage tail tip lysozyme, tail length tape measure protein, tail completion protein, Baseplate J-like protein, portal protein, and Mg2+/Co2+ transport protein (S4 Table).

Analysis of CRISPR loci in *L. buchneri* strains identified 27 spacers that target four human gut phages (BK020950, BK021565, BK038207, BK046275) sourced from NCBI's database (Fig 8C, S4 Table) [51,52]. Validation through IMG databases revealed that spacer 3 of the second CRISPR locus in strain 1012 specifically targets phage BK038207 (from the human gut), which encodes a tail terminator protein (TrP) [52]. The first spacer from the fourth CRISPR locus of strain S42, which targets BK021565, corresponded to phages found in bovine/sheep rumens and activated sludge, with the relevant area encoding a hypothetical protein that possesses an ABC-type ATPase domain and acetyltransferase, further indicated by Foldseek to include an AAA domain [50–52]. These results emphasize the role of CRISPR in the targeting of phages across various ecological contexts.

The genetic similarity among phages derived from human, bovine, sheep, and sludge sources ranged from 50.92% to 51.34% for human-animal pairs and from 51.02% to 51.17% for human-sludge pairs, with sludge phages exhibiting a high degree of intra-group similarity at 99.60%, in contrast to animal gut phages which showed a similarity of 71.27% (Fig 8C) [50–52]. Spacer 19 (strain S51) specifically targets phage BK020950 and a similar kefir phage with an 82.82% similarity, sharing conserved protospacer-PAM sequences that target a Sak4-like ssDNA annealing protein. Spacer 24 (LA1147) and spacer 6 (strain 1012) target different regions of the human gut phage BK046275: the portal protein and an unidentified hypothetical protein, respectively [50–52]. Even though the nucleotide similarity across phage sources is low (<52%), it is likely that CRISPR spacers recognize conserved functional domains (such as ATPase and portal proteins), highlighting the adaptability of CRISPR to divergent phages that possess essential genomic regions.

## Discussion

In this study, an in-depth comparative genomic exploration of forty *L. buchneri* strains isolated from tomato pulp, fermented cucumber, artisanal fermented pickles, fermented dough, silage, fermented sorghum product, cattle rumen, cheese, conjunctiva, fermented vegetables, fuel ethanol production facility, grape must, kimchi, and bovine nasopharynx was conducted. The genome sizes range between 2.33 and 2.76 Mb, which are consistent with those of lactic acid bacteria species (1.8 to 3.3 Mb). The average GC content achieved for *L. buchneri* was 44.18%, which is in alignment with low GC LAB. The sequence analysis of whole genomes of *L. buchneri* detected single or multiple plasmids in 17 out of 40 genomes, including the reference strain ATCC 4005. This further supports the hypothesis that LAB are well accustomed to their microniche by carrying plasmids in their genomes, which could swiftly be acquired and transmitted during rapid environmental condition changes [6].

We demonstrated the differences and similarities in phylogenetic trees based on the core and pangenomes of *L. buchneri*. The discrepancies seen across core versus pangenome-related phylogenetic trees, perhaps could be attributed to the accessory genome, including the presence of plasmids found in 42.5% of strains [9]. The discrepancies could also be attributed to inaccurate assemblies [53]. Pangenome outputs using Roary showed that *L. buchneri* has an open genome, suggesting the functional diversity of this species. The open pangenome permits and facilitates the retrieval of genetic components from outside environments to cope with adverse environmental conditions and adapt [54]. According to core orthogroups achieved in eggNOG-Mapper screening, a number of coding sequences were found to be associated with cell defense and repair, which are crucial for the growth and survival of microorganisms [55].

We detected a total of 72 intact prophages across all *L. buchneri* genomes with the exception of LA1175D and RUG14303, which belonged to fermented cucumber spoilage and cattle rumen microenvironments, respectively. Moreover, 22 genomic islands were identified according to fluctuating nucleotide sequence profiles. The existence of both mobile genetic elements of prophages and genomic islands implies the hallmark of the possible horizontal gene transfer events [6]. The percentage of unknown/hypothetical genes found in the whole genome was calculated to be ~23%, which suggests that there is room for discovery for functional evaluation studies of *L. buchneri*. With about ~31% of coding sequences identified that were conserved across 40 genomes screened, a significant amount of genetic diversity could be attributed to the accessory genome [56]. The existence of plasmids, genomic islands, and prophages proposed that mobile genetic elements are perhaps a significant genomic characteristic of *L. buchneri*.

Computational analysis of 40 *L. buchneri* genomes with regard to CRISPR-Cas systems using CRISPRviz and CRISPRCasFinder showed that all the strains encoded a putative CRISPR system. The abundance of CRISPR in this species is higher than that of lactobacilli and bacterial organisms overall, which suggests that *L. buchneri* is perhaps a remarkable reservoir for novel CRISPR-based tools [57]. Type-IIA was the most abundant CRISPR-Cas system found in *L. buchneri,* and this CRISPR type is the top candidate among all CRISPR systems for genetic engineering applications [58]. Type-IE CRISPR-Cas system was also represented in ~43% of strains, and this CRISPR toolbox can be reconfigured for genetic modification applications, especially in their native host [59].

Selective pressures within fermented environments (such as silage and kefir) influence the development of CRISPR-Cas systems in *L. buchneri*, evidenced by conserved patterns of spacers and direct repeats among different strains (Fig 7B) [60–63]. These common structures likely indicate an adaptation to phages and mobile genetic elements, with the duplication of spacers occurring through horizontal gene transfer, repeated acquisitions, or selective benefits [64]. The phylogenetic grouping of LbCas9 alongside distinct CRISPR profiles (Fig 7B) implies a coordinated evolutionary process, suggesting that these patterns could serve as potential phylogenetic indicators or tools for engineering strains resistant to phages [65,66]. Although the CRISPR adaptations in *L. buchneri* demonstrate ecological specialization within fermented environments, it remains essential to conduct functional validation for industrial use [67,68].

*L. buchneri*'s CRISPR spacers are effective against phages found in various environments, including the human gut microbiome, kefir, the rumen of livestock, and sewage (Fig 9) [69–72]. Its utilization as a silage inoculant [67,68] and its presence in crops such as tomatoes and cucumbers indicate ecological connections, likely facilitated by the use of manure fertilizers that transfer bacteria to soil and plants [73,74]. Although existing databases tend to emphasize phages from the human gut, the independent targeting of similar phages by different strains (for example, from cucumber and fermented dough) suggests significant CRISPR-phage interactions [75]. This highlights *L. buchneri*'s ability to adapt across various niches and the extensive defense capacity of its CRISPR system, which is relevant for food safety and microbial ecology.

Although the protospacer-spacer matches were statistically significant (e-value ≤ 0.001) for the 5 spacers that are 30 nts in length and putatively targeting the phages, there were up to three mismatches across 4 of the protospacers and a single PAM with a mismatch in the 5'-AAAA-3' core (Fig 8C, S4 Table). For Cas9, the mismatches between spacer-protospacer that are neither in strongly conserved parts of PAM nor in the seed region (~10 nts upstream of PAM) are well tolerated [76]. The mismatches could be due to: i) natural variations in phage strains due to genetic drift, ii) mutations in phage genomes that are positively selected under the CRISPR pressure, iii) the prophage stage of the life-cycle that needs to be refractory to dsDNA breaks due to lethality in the host [6]. It is likely possible that in the future, as more phages are sequenced and deposited to the databases, even stronger matches could be identified.

Because *L. buchneri* can also be detrimental to food bioprocessing due to causing food spoilage, particularly in the fermented cucumber industry, it is crucial to control the contamination of this organism to eliminate defects caused by this species [77,78]. In-depth CRISPR spacer profiling could be used to eliminate the potential contamination of *L. buchneri* by designing phage therapy [6]. While phage therapy presents a promising strategy to control *L. buchneri*-induced spoilage

 

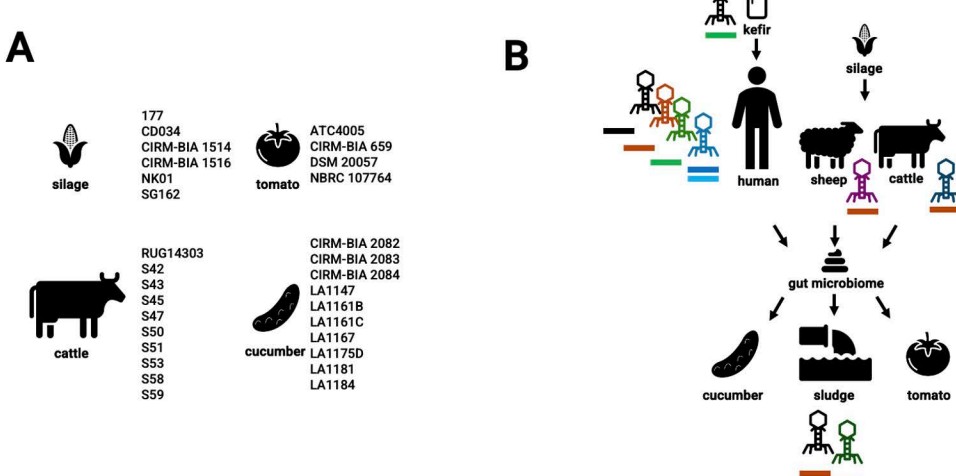

**Fig 9.** (A) **Categorization of *L. buchneri* strains based on their sources of isolation.** (B) Representative schematic of the potential lifecycle of *L. buchneri* strains starting as part of starter cultures used in kefir and silage. Bacteriophages that are targeted by CRISPR resistome are indicated by their sources of isolation. Vertical colored bars below phages represent spacer numbers and identities targeting each phage. (Phage, cucumber, tomato, stool, and sewage icons are sourced from flaticon.com).

in fermented foods, its ethical and environmental implications warrant careful consideration. Potential off-target effects on beneficial microbes, ecological disruptions, and the risk of horizontal gene transfer must be assessed [79,80]. The use of CRISPR spacer data to guide phage design may also raise concerns about accelerating microbial resistance dynamics [81]. Responsible application requires risk assessment, regulatory compliance, and transparency to ensure safe integration into food systems [82].

The extensive genome evaluation of *L. buchneri* revealed that this species carries genes as they pertain to CAZymes, which are instrumental in carbohydrate synthesis and hydrolysis during fermentation. Although glycoside hydrolases, carbohydrate esterases, auxiliary activities, and carbohydrate-binding modules were associated with degradation reactions, glycoside transferases participate in carbohydrate biosynthesis [83]. The abundance of GH-related genes found in *L. buchneri* genomes suggests the carbohydrate fermentation capability of *L. buchneri* since sugar utilization is a significant demarcation of a bacterium's functionality and creates a baseline for strain cultivation and selection [84].

It has been reported that one of the most frequently encountered amine-producing bacteria present in sufficient quantities in dairy products is *L. buchneri*. For example, *L. buchneri* was found in Swiss cheese that had a significant concentration of histamine. Strains of *L. buchneri* isolated from Gouda cheese were responsible for the production of histamine and tyramine, respectively [85], implying a potential mechanism of histidine to histamine conversion. In contrast, our genome survey detected no complete hdcA-B-C cassette among the 40 publicly available strains, suggesting that histamine potential is uncommon at the species level and likely restricted to rare lineages. Histidine decarboxylase, functional in the catabolism of histamine from histidine, was only found in LA1181, which was isolated from reduced NaCl fermented cucumber spoilage. However, the remaining two sets of genes (i.e., *hdcB* and *hdcC*) were missing, which might be due to evolutionary gene loss, perhaps as a result of long proliferation of strain LA1181 under histidine-limited conditions of fermented cucumber [70].

## Conclusion

The present study set out to assess strain-level biodiversity within *Lentilactobacillus buchneri* by analyzing 40 complete genomes and to chart the repertoire of endogenous CRISPR-Cas loci with a view to exploiting—or controlling—this

species in food bioprocesses. Phylogenomic reconstruction revealed marked intra-species diversity, while principal-coordinates analysis showed no consistent clustering by isolation source. Together with the nearly identical carbohydrate active enzyme profiles observed across all origins, this pattern supports a predominantly free-living lifestyle for *L. buchneri*. The high prevalence of intact prophages and plasmids in most genomes further underscores their genomic plasticity. Contrary to long-standing concerns about histamine formation, none of the 40 genomes carried a complete histidine-decarboxylase gene cassette, emphasizing the need for strain-specific functional assays to verify histamine-forming potential in vitro. Detailed inspection of CRISPR arrays and cas genes reveals ongoing co-evolution with bacteriophages that share the same ecological niches. Deciphering these CRISPR-mediated immune responses provides a foundation for future biotechnological applications, ranging from endogenous "CRISPRization" of *L. buchneri* strains to rationally designed phage interventions.

## Supporting information

**S1 Table. CRISPRCasFinder results of *L. buchneri* genomes tested.**
(PDF)

**S2 Table. Putative prophages predicted in *L. buchneri* strains using PHASTEST.**
(PDF)

**S3 Table. Putative plasmids found in *L. buchneri* strains.**
(PDF)

**S4 Table. Megablast search results showing best CDS hits targeted by spacers.**
(PDF)

**S1 Fig. BUSCO assessment results of 46 *L. buchneri* genome assemblies.**
(TIF)

**S2 Fig. Alignment of spacers (A) and repeats (B) of each detected CRISPR locus.** Each colored diamond represents a unique repeat, and each colored square represents a unique spacer in the CRISPR-Cas system. Grey "x" boxes showed a missing spacer.
(TIF)

**S3 Fig. Unrooted phylogenetic tree of 33 LbCas9s based on primary amino acid sequence similarity.**
(TIF)

## Acknowledgments

Not applicable.

## Author contributions

**Conceptualization:** Ibrahim Cagri Kurt, Fatih Ortakci.

**Data curation:** Ismail Gumustop, Ibrahim Genel, Ibrahim Cagri Kurt, Fatih Ortakci.

**Formal analysis:** Ismail Gumustop, Ibrahim Genel, Fatih Ortakci.

**Funding acquisition:** Fatih Ortakci.

**Investigation:** Ismail Gumustop, Ibrahim Genel, Ibrahim Cagri Kurt, Fatih Ortakci.

**Methodology:** Ismail Gumustop, Ibrahim Cagri Kurt, Fatih Ortakci.

**Project administration:** Fatih Ortakci.

**Resources:** Fatih Ortakci.

**Software:** Fatih Ortakci.

**Supervision:** Ibrahim Cagri Kurt, Fatih Ortakci.

**Validation:** Ismail Gumustop, Ibrahim Cagri Kurt, Fatih Ortakci.

**Visualization:** Ismail Gumustop, Ibrahim Genel, Ibrahim Cagri Kurt, Fatih Ortakci.

**Writing – original draft:** Ismail Gumustop, Ibrahim Genel, Ibrahim Cagri Kurt, Fatih Ortakci.

**Writing – review & editing:** Ibrahim Cagri Kurt, Fatih Ortakci.

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
