## [Decision Letter · Decision Letter 0]

Dear Dr. ORTAKCI,

Dear Authors 

We look forward to receiving your revised manuscript.

Kind regards,

Aziz ur Rahman Muhammad

Academic Editor

PLOS ONE

“This work has been supported by the Istanbul Technical University Scientific Research Projects Unit with grant number MGA-2023-45115.”

Additional Editor Comments:

Dear Authors

Thanks for submitting innovative work in our Journal. However, reviewers suggested major revision before publication of the manuscirpt. Therefore, i invite you to revise the manuscirpt as suggested by reviewers. Please make sure manuscript should be formatted according to Journal requirements. For example, manuscirpt abstract is lengthy please revise according to suggested guidelines of the Journals. Manuscirpt is too long, please provide only important information in it. Increase the quality of figures especially supplementary materials figures.

Reviewers' comments:

Reviewer's Responses to Questions

**Comments to the Author**

1. Is the manuscript technically sound, and do the data support the conclusions?

Reviewer #1: Yes

Reviewer #2: Yes

Reviewer #3: Yes

2. Has the statistical analysis been performed appropriately and rigorously?

Reviewer #1: No

Reviewer #2: N/A

Reviewer #3: N/A

3. Have the authors made all data underlying the findings in their manuscript fully available?

Reviewer #1: Yes

Reviewer #2: Yes

Reviewer #3: Yes

4. Is the manuscript presented in an intelligible fashion and written in standard English?

Reviewer #1: No

Reviewer #2: Yes

Reviewer #3: Yes

Reviewer #1: Dear author, i am pleased to see your work and interest. You have explained the results in bioinformatics tools with attractive pictorial figures. Manuscript is mainly written in theory pattern and at many sections, veery large paras are put even some where no results' presentation was shown. Work is valuable and i suggest need little scientific and major literally improvement.

Reviewer #2: The authors attempt to demonstrate by bioinformatics and molecular methods support the hypothesis the diversity at the species level and uncovering the biotechnological potential for better use of this species at fermentation bioprocessing applications.

The work is important because the main focus is on the genome sequencing advancements, annotation and validation through in-silico techniques, to find Comprehensive analysis of the CRISPR resistome reflects the co-evolutionary history of L. buchneri and bacteriophages that

co-occupy the natural lifecycle and habitats of L. buchneri. The insights gained from L.buchneri CRISPR immunity enable potential biotechnological applications of CRISPRization and phage therapy.The title reflects the finding of the manuscript.

The abstract should be shorter, between 250 and 300 words, it gets a bit lost as it is integrated.

The manuscript is for the most part well written and shows robust methods, though some grammatical corrections will be necessary, and the use of prepositions should be reviewed. Also please make sure that the name of the bacteria is spelled correctly and in italics.

Line 70 pseudomonas change by Pseudomonas in italic

Line 71 lactobacillus, lactococcus, leuconostoc change by Lactobacillus, Lactococcus in italic

Line 72 streptococcus, and enterococcus change by Streptococcus and Enterococcus in italic

The Whole-genome sequences of 46 L. buchneri strains were downloaded from NCBI should be a table.

If you relate the figure numbers all must be written in bold. For example

Line 219 exist in pangenome that were annotated with Prokka which is shown in Fig 3. Change by exist in pangenome that were annotated with Prokka which is shown in Fig 3.

Line 240 core genome (Fig. 5A). change by core genome (Fig. 5A).

The most interesting conclusion is the one they literally wrote down Athorough examination of the CRISPR defense system reveals the co-evolutionary relationship between L. buchneri and the bacteriophages that share its natural environments and lifecycle.Understanding L. buchneri's CRISPR-based immune responses opens up possibilities for biotechnological applications, including CRISPR manipulation and phage therapy. The CRISPR system has opened up several research fronts that should be used to solve biotechnological and therapeutic problems. However, the ethical aspects of this should be considered.

Check the citations as they are not written in a homogeneous way, as some have doi, others do not. Also, the names of bacteria should be italicized. For example,

The authors wrote Nethery MA, Henriksen ED, Daughtry KV, Johanningsmeier SD, Barrangou R. Comparative genomics of eight Lactobacillus buchneri strains isolated from food spoilage. BMC Genomics. 2019; 20:902.

Cite AMA Nethery MA, Henriksen ED, Daughtry KV, Johanningsmeier SD, Barrangou R. Comparative genomics of eight Lactobacillus buchneri strains isolated from food spoilage. BMC Genomics. 2019;20(1):902. Published 2019 Nov 27. doi:10.1186/s12864-019-6274-0

The authors wrote Mojica FJM, Díez-Villaseñor C, García-Martínez J, Almendros C. Short motif sequences determine the targets of the prokaryotic CRISPR defence system. Microbiology. Microbiology Society; 2009. p. 733–40.

Cite AMA Mojica FJM, Díez-Villaseñor C, García-Martínez J, Almendros C. Short motif sequences determine the targets of the prokaryotic CRISPR defence system. Microbiology (Reading). 2009;155(Pt 3):733-740. doi:10.1099/mic.0.023960-0

The figures need to be improved as in some of them the pixel’s are not visible. For example, the figure 4

the work is interesting, however it is too long, they should try to shorten some parts as the reader will get tired of reading it.

Reviewer #3: The manuscript "Comparative Genomics of Lentilactobacillus buchneri Reveals Strain Level Hyperdiversity and Broad-spectrum CRISPR Immunity Against Human and Livestock Gut Phages"represents analysis of 40 L. buchneri genomes available in the GenBank database.

It is thoroughly performed and technically sound bioinformatic study, it reveals common genomic features, genetic peculiarities, and CRIPR systems of L. buchneri strains. At the same time, this manuscript should be improved, as it contains some inaccuracies and unclear ideas.

Comments to the authors:

1. Abstract is too large (do not exceed 300 words). According to the Submission Guidelines, the Abstract should:

• Describe the main objective(s) of the study

• Explain how the study was done, including any model organisms used, without methodological detail

• Summarize the most important results and their significance

• Not exceed 300 words

2. Lines 59-60 “...grows at 15 °C but no growth occurs at 45 °C”. L. buchneri grows in a wide range of temperatures (15-37C).

3. Lines 91-93 “…..uncovering its metabolic potentials in addition to strain level differences. To better understand …. biotechnological traits of L. buchneri…” This goal poorly correlates with the results and the title of the manuscript, as whole genome parameters, differences, and CRISPR systems are mainly described, but not metabolic potential and biotechnological traits.

4. Line 172:

- “Forty-six L. buchneri genomes were downloaded from NCBI GenBank [2]”

At least, 49 genomes of L. buchneri strains are available in NCBI (https://www.ncbi.nlm.nih.gov/datasets/genome/?taxon=1581). Please, clarify, why only 46/40 genomes were selected for this study.

- Reference [2] is not necessary.

5. Table 1. Strains S42-S59 were isolated in the same year (2014) and in the same place (Canada:Lethbridge) (https://www.ncbi.nlm.nih.gov/bioproject/533291). In addition, they demonstrate very high level of intergenomic similarity, with the exception of S51 (Figure 1, Figure 2). Probably, they are isolates of one strain circulating in bovine nasopharynx in this particular farm. This data affects the results of the study and should be noted in the text and discussed.

6. Section “Analysis of Carbohydrate Active Enzymes” should be re-written. Lines 266-277 should be first, as it is a common data for genomes annotation. The authors did not find complete hdc clusters in all 40 analyzed genomes, is there any literature evidence that this cluster was detected in L. buchneri earlier?

7. Line 306. “…the highest number of intact prophage…” Needs plural number.

8. Lines 391-395. The sentence is not clear. Please, shorten it or re-written.

9. Lines 396-407. Please, clarify the idea to analyze these putative phages for their genome similarity. According to your analysis, they have a very low nucleotide similarity. Do you have any conclusions from these calculations?

10. Lines 459-509. A very long and incomprehensible fragment. Please shorten and clarify your ideas. Considering that CRISPR spacers are very short (30 nt) and natural phages are very diverse, these assumptions are highly speculative and require further study. In addition, human intestinal virome is the most studied community of phages, so it is easiest to find similar sequences in it.

It was impossible to download Figures and Supplementary files. Therefore, I have no opportunity to review Supplementary.

**Do you want your identity to be public for this peer review?** For information about this choice, including consent withdrawal, please see our Privacy Policy

Reviewer #1: **Yes: ** Dr. Riaz Mustafa

Reviewer #2: No

Reviewer #3: No

---

## [Author Response · Author response to Decision Letter 1]

2 May 2025

Response to Reviewer Comments

Reviewer #1: Dear author, i am pleased to see your work and interest. You have explained the results in bioinformatics tools with attractive pictorial figures. Manuscript is mainly written in theory pattern and at many sections, veery large paras are put even some where no results' presentation was shown. Work is valuable and i suggest need little scientific and major literally improvement.

AU: Thank you for the constructive and insightful comments. We conducted a thorough revision of manuscript and summarized the paragraphs into shorter versions with sound improvements made to show concise results achieved related to the analysis performed.

Reviewer #2: The authors attempt to demonstrate by bioinformatics and molecular methods support the hypothesis the diversity at the species level and uncovering the biotechnological potential for better use of this species at fermentation bioprocessing applications.

The work is important because the main focus is on the genome sequencing advancements, annotation and validation through in-silico techniques, to find Comprehensive analysis of the CRISPR resistome reflects the co-evolutionary history of L. buchneri and bacteriophages that

co-occupy the natural lifecycle and habitats of L. buchneri. The insights gained from L.buchneri CRISPR immunity enable potential biotechnological applications of CRISPRization and phage therapy.The title reflects the finding of the manuscript.

The abstract should be shorter, between 250 and 300 words, it gets a bit lost as it is integrated.

AU: Thank you for the kind comments and appreciation for our work. We revised the abstract and made it to the point and it is now 250>x>300 words.

The manuscript is for the most part well written and shows robust methods, though some grammatical corrections will be necessary, and the use of prepositions should be reviewed. Also please make sure that the name of the bacteria is spelled correctly and in italics.

AU: The manuscript has been through a thorough grammar and preposition check and necessary corrections/revisions made accordingly. Bacterial names were italicized accordingly.

Line 70 pseudomonas change by Pseudomonas in italic

Line 71 lactobacillus, lactococcus, leuconostoc change by Lactobacillus, Lactococcus in italic

Line 72 streptococcus, and enterococcus change by Streptococcus and Enterococcus in italic

AU: Revised accordingly.

The Whole-genome sequences of 46 L. buchneri strains were downloaded from NCBI should be a table.

AU: Thank you for the comment. Table 1 shows the whole genome sequence statistics of 40 genomes that passed the BUSCO quality check with regards to accession numbers, isolation sources, size etc.

If you relate the figure numbers all must be written in bold. For example

Line 219 exist in pangenome that were annotated with Prokka which is shown in Fig 3. Change by exist in pangenome that were annotated with Prokka which is shown in Fig 3.

Line 240 core genome (Fig. 5A). change by core genome (Fig. 5A).

AU: Revised accordingly

The most interesting conclusion is the one they literally wrote down Athorough examination of the CRISPR defense system reveals the co-evolutionary relationship between L. buchneri and the bacteriophages that share its natural environments and lifecycle.Understanding L. buchneri's CRISPR-based immune responses opens up possibilities for biotechnological applications, including CRISPR manipulation and phage therapy. The CRISPR system has opened up several research fronts that should be used to solve biotechnological and therapeutic problems. However, the ethical aspects of this should be considered.

AU: Thank you for the insightful comment. The end of the discussion section was revised accounting the ethical aspects of CRISPR applications with proper references added from the literature.

Check the citations as they are not written in a homogeneous way, as some have doi, others do not. Also, the names of bacteria should be italicized. For example,

The authors wrote Nethery MA, Henriksen ED, Daughtry KV, Johanningsmeier SD, Barrangou R. Comparative genomics of eight Lactobacillus buchneri strains isolated from food spoilage. BMC Genomics. 2019; 20:902.

Cite AMA Nethery MA, Henriksen ED, Daughtry KV, Johanningsmeier SD, Barrangou R. Comparative genomics of eight Lactobacillus buchneri strains isolated from food spoilage. BMC Genomics. 2019;20(1):902. Published 2019 Nov 27. doi:10.1186/s12864-019-6274-0

The authors wrote Mojica FJM, Díez-Villaseñor C, García-Martínez J, Almendros C. Short motif sequences determine the targets of the prokaryotic CRISPR defence system. Microbiology. Microbiology Society; 2009. p. 733–40.

Cite AMA Mojica FJM, Díez-Villaseñor C, García-Martínez J, Almendros C. Short motif sequences determine the targets of the prokaryotic CRISPR defence system. Microbiology (Reading). 2009;155(Pt 3):733-740. doi:10.1099/mic.0.023960-0

The figures need to be improved as in some of them the pixel’s are not visible. For example, the figure 4

AU: Thank you for the detailed feedback on citation formats. We revised them accordingly. And Figure 4 quality has been improved.

the work is interesting, however it is too long, they should try to shorten some parts as the reader will get tired of reading it.

AU: Thank you for the comment. The lengthy paragraphs were either trimmed or broken into multiple pieces to ease up the tracking of the manuscript and to improve clarity.

Reviewer #3: The manuscript "Comparative Genomics of Lentilactobacillus buchneri Reveals Strain Level Hyperdiversity and Broad-spectrum CRISPR Immunity Against Human and Livestock Gut Phages"represents analysis of 40 L. buchneri genomes available in the GenBank database.

It is thoroughly performed and technically sound bioinformatic study, it reveals common genomic features, genetic peculiarities, and CRIPR systems of L. buchneri strains. At the same time, this manuscript should be improved, as it contains some inaccuracies and unclear ideas.

Comments to the authors:

1. Abstract is too large (do not exceed 300 words). According to the Submission Guidelines, the Abstract should:

• Describe the main objective(s) of the study

• Explain how the study was done, including any model organisms used, without methodological detail

• Summarize the most important results and their significance

• Not exceed 300 words

AU: Thank you for the comments. The abstract was revised accordingly.

2. Lines 59-60 “...grows at 15 °C but no growth occurs at 45 °C”. L. buchneri grows in a wide range of temperatures (15-37C).

AU: Revised accordingly.

3. Lines 91-93 “…..uncovering its metabolic potentials in addition to strain level differences. To better understand …. biotechnological traits of L. buchneri…” This goal poorly correlates with the results and the title of the manuscript, as whole genome parameters, differences, and CRISPR systems are mainly described, but not metabolic potential and biotechnological traits.

AU: Thank you for the great catch. The relevant paragraph has been revised accordingly to better correlate with the result and title of the manuscript.

4. Line 172:

- “Forty-six L. buchneri genomes were downloaded from NCBI GenBank [2]”

At least, 49 genomes of L. buchneri strains are available in NCBI (https://www.ncbi.nlm.nih.gov/datasets/genome/?taxon=1581). Please, clarify, why only 46/40 genomes were selected for this study.

AU: The discrepancy between the number of L. buchneri genomes available in NCBI (currently ≥49) and those analyzed in the study arises from strict quality filtering criteria applied during genome selection. The selection of 40 genomes reflects a methodological focus on quality and comparability, not an oversight. We prioritized high-confidence genomic data to ensure reliable conclusions about CRISPR systems, prophages, and strain diversity. The remaining 6 genomes (from the initial 46) were excluded due to BUSCO failures while newer NCBI entries (post-study) account for the current total of ≥49.This approach is standard in genomics to avoid biases from incomplete or misassembled data. The study’s conclusions remain valid for the rigorously curated subset analyzed.

Our genomic study relies on a fixed dataset extracted from NCBI at the time of analysis to ensure reproducibility and methodological consistency. We downloaded genomes before newer entries were uploaded, as NCBI is continuously updated. The lag between data collection, manuscript preparation, peer review, and publication (often months to years) means newer genomes submitted during this period are excluded. This is unavoidable in rapidly evolving fields like genomics. Re-running analyses to include newer genomes post-submission is often impractical due to computational/resource limitations and the need to finalize findings for publication.

The exclusion of newer genomes is not a methodological flaw but a reflection of the dynamic nature of genomic databases. The study’s conclusions remain valid for the dataset analyzed, and future work can build on these findings by incorporating newer entries. This approach ensures transparency and replicability, as readers can verify results against the original dataset (40 genomes) used in the study.

- Reference [2] is not necessary.

AU:Revised accordingly.

5. Table 1. Strains S42-S59 were isolated in the same year (2014) and in the same place (Canada:Lethbridge) (https://www.ncbi.nlm.nih.gov/bioproject/533291). In addition, they demonstrate very high level of intergenomic similarity, with the exception of S51 (Figure 1, Figure 2). Probably, they are isolates of one strain circulating in bovine nasopharynx in this particular farm. This data affects the results of the study and should be noted in the text and discussed.

AU: Thank you for the great comment. According to the ANI calculations (see below) not a 100% hit was received between S42 thru S59 implying that these strains are unique. We agree that the location of the isolates and dates isolated could perhaps mean that there might be genetic drift seen from a single isolate which was mentioned and discussed in the manuscript accordingly.

ANI Scores (%)

S42

S43

S45

S47

S50

S51

S53

S58

S59

S42

100

99,97

99,97

99,97

99,97

97,61

99,96

99,96

99,97

S43

99,97

100

99,97

99,97

99,97

97,61

99,97

99,97

99,97

S45

99,97

99,97

100

99,97

99,97

97,59

99,97

99,97

99,97

S47

99,97

99,97

99,97

100

99,97

97,62

99,97

99,97

99,97

S50

99,97

99,97

99,97

99,97

100

97,58

99,97

99,97

99,97

S51

97,61

97,61

97,59

97,62

97,58

100

97,61

97,61

97,58

S53

99,96

99,97

99,97

99,97

99,97

97,61

100

99,97

99,97

S58

99,96

99,97

99,97

99,97

99,97

97,61

99,97

100

99,97

S59

99,97

99,97

99,97

99,97

99,97

97,58

99,97

99,97

100

SG162

99,85

99,85

99,79

99,86

99,78

97,62

99,85

99,85

99,78

6. Section “Analysis of Carbohydrate Active Enzymes” should be re-written. Lines 266-277 should be first, as it is a common data for genomes annotation. The authors did not find complete hdc clusters in all 40 analyzed genomes, is there any literature evidence that this cluster was detected in L. buchneri earlier?

AU: Carbohydrate Active Enzyme part was rewritten accordingly. And eggNog annotations were brought earlier in the text. Thank you for the insightful comment on histidine decarboxylase gene. We added a paragraph to the discussion section from the literature (Calasso and Gobetti 2011-https://www.sciencedirect.com/topics/immunology-and-microbiology/lactobacillus-buchneri) showing that L. buchneri is responsible for histamine production from histidine in multiple aging cheese varieties. Therefore, we wanted to screen L. buchneri genomes for the presence of hdc genes.

7. Line 306. “…the highest number of intact prophage…” Needs plural number.

AU:Revised accordingly.

8. Lines 391-395. The sentence is not clear. Please, shorten it or re-written.

AU:Revised accordingly

9. Lines 396-407. Please, clarify the idea to analyze these putative phages for their genome similarity. According to your analysis, they have a very low nucleotide similarity. Do you have any conclusions from these calculations?

AU: Thank you for the comment. The text was revised to provide more clarity and raw conclusions. We added the following statement ‘’The low nucleotide similarity (<52%) between phages from human, animal, and environmental sources implies that CRISPR spacers in these strains might recognize conserved functional domains (e.g., ATPase, portal proteins) in addition to closely related phage genomes. This highlights the adaptability of CRISPR systems to target divergent phages sharing critical genomic regions.’’

10. Lines 459-509. A very long and incomprehensible fragment. Please shorten and clarify your ideas. Considering that CRISPR spacers are very short (30 nt) and natural phages are very diverse, these assumptions are highly speculative and require further study. In addition, human intestinal virome is the most studied community of phages, so it is easiest to find similar sequences in it.

AU: The text has been revised to address the ambiguity and the length.

---

## [Decision Letter · Decision Letter 1]

Comparative Genomics of Lentilactobacillus buchneri Reveals Strain Level Hyperdiversity and Broad-spectrum CRISPR Immunity Against Human and Livestock Gut Phages

PONE-D-25-08247R1

Dear Dr. ORTAKCI,

We’re pleased to inform you that your manuscript has been judged scientifically suitable for publication and will be formally accepted for publication once it meets all outstanding technical requirements.

Kind regards,

Aziz ur Rahman Muhammad

Academic Editor

PLOS ONE

Additional Editor Comments (optional):

Dear Authors

Thanks for revision

Reviewers' comments:

Reviewer's Responses to Questions

**Comments to the Author**

Reviewer #1: All comments have been addressed

Reviewer #2: All comments have been addressed

2. Is the manuscript technically sound, and do the data support the conclusions?

Reviewer #1: Yes

Reviewer #2: Yes

3. Has the statistical analysis been performed appropriately and rigorously?

Reviewer #1: Yes

Reviewer #2: N/A

4. Have the authors made all data underlying the findings in their manuscript fully available?

Reviewer #1: Yes

Reviewer #2: Yes

5. Is the manuscript presented in an intelligible fashion and written in standard English?

Reviewer #1: Yes

Reviewer #2: Yes

Reviewer #1: Good work and improvement and appreciate the author's work and his team for this in silico work. I hope the author will continue this struggle for his future avenues

Reviewer #2: The authors have done an excellent job and have responded point by point to the reviewers' comments. In addition, it is a work with the scientific rigour required for publication.

**Do you want your identity to be public for this peer review?** For information about this choice, including consent withdrawal, please see our Privacy Policy

Reviewer #1: **Yes: ** Riaz Mustafa

Reviewer #2: **Yes: ** Graciela Castro-Escarpulli Escuela Nacional de Ciencias Biológicas Instituto Politécnico Nacional México

---

## [Editor Report · Acceptance letter]

PONE-D-25-08247R1

PLOS ONE

Dear Dr. ORTAKCI,

I'm pleased to inform you that your manuscript has been deemed suitable for publication in PLOS ONE. Congratulations! Your manuscript is now being handed over to our production team.

Kind regards,

on behalf of

Dr. Aziz ur Rahman Muhammad

Academic Editor

PLOS ONE